# Boosting Broiler Health and Productivity: The Impact of in ovo Probiotics and Early Posthatch Feeding with *Bacillus subtilis*, *Lactobacillus fermentum*, and *Enterococcus faecium*

**DOI:** 10.3390/microorganisms13061219

**Published:** 2025-05-27

**Authors:** Jan P. Madej, Anna Woźniak-Biel, Andrzej Gaweł, Kamila Bobrek

**Affiliations:** 1Department of Immunology, Pathophysiology and Veterinary Preventive Medicine, Wrocław University of Environmental and Life Sciences, 50-375 Wrocław, Poland; jan.madej@upwr.edu.pl; 2Department of Epizootiology and Clinic of Birds and Exotic Animals, Wrocław University of Environmental and Life Sciences, 50-366 Wrocław, Poland; anna.wozniak-biel@upwr.edu.pl (A.W.-B.); andrzej.gawel@upwr.edu.pl (A.G.)

**Keywords:** probiotic, in ovo, early feeding, chicken, intestine, *Campylobacter*, immune organs, *Bacillus subtilis*, *Lactobacillus fermentum*, *Enterococcus faecium*

## Abstract

In ovo administration of probiotics has the potential to enable the early colonization of the gut microbiota, providing health benefits from the onset of life. This study aimed to assess the impact of in ovo probiotic inoculation combined with early posthatch feeding on intestinal development and colonization by *Campylobacter* spp., immune system development, and the final production performance of chickens. On the 18th day of incubation, *Bacillus subtilis*, *Lactobacillus fermentum*, *Enterococcus faecium*, or physiological saline (control) was administered to Ross 308 eggs in ovo, and chicks had immediate access to feed and water upon hatching. On the 7th, 21st, and 35th days after hatching, samples of tissues were taken for histomorphometric analysis. *Campylobacter* strains in the cecal content were quantitatively evaluated. Probiotic administration had a beneficial effect on the development of the small intestine and increased the number of B cells in the spleen and the number of B and CD4^+^ cells in the cecal tonsils. The in ovo administration of probiotics did not reduce *Campylobacter jejuni* colonization and even led to increased bacterial loads in some groups by day 35. However, when combined with early feeding, in ovo probiotic administration had a positive impact on the development of the small intestine and peripheral immune organs.

## 1. Introduction

The global poultry industry is a cornerstone of agricultural production, supplying a significant portion of the world’s protein intake through broiler chicken meat. Ensuring the health and productivity of broiler chickens is paramount, as these factors directly influence economic viability and food security. Traditional reliance on antibiotic growth promoters (AGPs) to enhance growth performance and prevent diseases has raised concerns due to the emergence of antibiotic-resistant bacteria and potential residues in meat products [1,2,3,4,5]. Consequently, there is a pressing need to explore alternative strategies that promote poultry health and productivity without compromising food safety or public health. Probiotics have emerged as a promising alternative to AGPs, offering improved gut health, enhanced immune responses, and better growth performance [6,7,8]. In particular, the in ovo administration of probiotics—injecting beneficial microorganisms into the egg before hatching—has garnered attention for its potential to enable the early colonization of the gut microbiota, thereby conferring health benefits from the onset of life. In ovo probiotic administration aims to provide chicks with a competitive advantage against pathogenic microorganisms immediately after hatching, promoting optimal development and performance [9]. The in ovo method, while promising, faces several challenges that may impact its effectiveness. One major issue is the variability in substance absorption, which depends on the injection site and the timing during embryonic development [10]. Additionally, technical aspects of the application, along with concerns about probiotic survival rates and their uniform distribution, may limit the method’s use in large-scale poultry production. Despite certain limitations in the application of probiotics via in ovo methods, this technique remains promising for future poultry production due to its potential benefits. Among probiotic strains, *B. subtilis*, *L. fermentum*, and *E. faecium* have demonstrated notable effects in poultry. *Bacillus subtilis* is a spore-forming bacterium known for its resilience and ability to produce enzymes that aid in nutrient digestion [11]. Studies have indicated that *Bacillus subtilis* supplementation can improve growth performance and modulate the gut microbiota in broilers. For instance, a study by Mohamed et al. [12] demonstrated that the dietary inclusion of *Bacillus subtilis* ATCC19659 as a feed additive positively affected the growth performance, immune response, and cecal microbiota of broiler chickens. Similarly, *Lactobacillus fermentum*, a lactic acid bacterium, has been associated with improved gut health and immune modulation. Penha Filho et al. [13] reported that *Lactobacillus fermentum* administration modulated immune responses and reduced pathogen colonization in chickens. *Enterococcus faecium* has also been studied for its probiotic properties, with findings suggesting benefits in growth performance and gut health. Castañeda et al. [14] found that *Enterococcus faecium* supplementation improved the absorption of nutrients and early immune system maturation in broilers. The in ovo administration of these probiotics offers a unique approach to modulating the gut environment from the earliest stages of development. Introducing beneficial bacteria into the egg can shape the gut microbiota, enhance immune system maturation, and improve nutrient absorption posthatch [14]. Despite the promising potential of in ovo probiotic administration, several factors must be considered to optimize its efficacy. The selection of probiotic strains, their viability during the in ovo process, the timing and method of administration, and the dosage are critical parameters that influence outcomes. One of the primary benefits of probiotics is their ability to competitively exclude avian and human pathogens. However, the effectiveness of probiotic strategies can vary significantly depending on several factors, including the chicken breed, the specific probiotic strain used, the timing of administration, and the method of delivery, whether through feed, water, spray, or in ovo injection [15]. Although early-life applications tend to yield more favorable outcomes, some interventions have been shown to disrupt the existing gut microbiota, potentially leading to unintended microbial imbalances [16]. Moreover, in some instances, probiotics have failed to significantly reduce *Campylobacter* spp., underscoring the need for more targeted and evidence-based intervention strategies [17,18,19]. *Campylobacter* species, particularly *C. jejuni* and *C. coli*, are the leading causes of bacterial gastroenteritis worldwide, posing significant public health challenges [20]. Infections, often resulting from consuming contaminated food or water, manifest as symptoms such as abdominal pain, diarrhea, and fever [21]. While many cases are self-limiting, severe complications such as Guillain–Barré syndrome and reactive arthritis can occur [22]. The global incidence of campylobacteriosis has been rising, with notable increases reported in both developed and developing countries [21]. Poultry and poultry products are major sources of human infections, highlighting the need for effective intervention measures along the food chain to prevent and control transmission [20]. Early feeding in broiler chicks has been shown to enhance gastrointestinal development and immune function. Providing immediate access to feed and water posthatch stimulates the development of gut-associated lymphoid tissue, contributing to improved immune responses [23]. Additionally, early feeding promotes the development and maturation of gastrointestinal tract microbiota, leading to better nutrient absorption and overall health [24]. Our previous studies indicated that the early posthatch feeding method not only improves the production parameters of broilers but also increases their welfare [23,25]. In the present study, it was hypothesized that early posthatch feeding combined with in ovo delivery of selected probiotics would have a prolonged positive impact on production parameters and on the development and function of the intestine and immune system. This study also aimed to evaluate whether the selected in ovo -administered probiotics would reduce the shedding of *Campylobacter* spp. by chicks.

## 2. Materials and Methods

### 2.1. Probiotics

The probiotics used in this experiment—*Bacillus subtilis, Lactobacillus fermentum*, and *Enterococcus faecium*—were provided by BIO GEN (Przedsiębiorstwo Wdrożeń i Zastosowań Biotechnologii i Inżynierii Genetycznej BIO GEN SP. Z O.O, Namyslow, Poland) in a lyophilized form. The *Bacillus subtilis* strain was resistant only to colistin, *Enterococcus faecium* to amoxicillin, enrofloxacin, doxycycline, tetracycline, colistin sulfonamides, and florfenicol, and *Lactobacillus fermentum* to enrofloxacin, tetracycline, colistin, and sulfonamides. The lyophilized strains were cultured in Brain–Heart Infusion (BHI) broth (Oxoid, Hampshire, UK) and incubated for 24 h at 37 °C. PCR was performed to confirm the probiotic genera and species. DNA was extracted using the Genomic Mini Kit (A&A Biotechnology, Gdynia, Poland) according to the manufacturer’s instructions. The PCR reactions were performed in a 25 µL reaction mixture containing 5 ng of template DNA, 10 pmol of each primer (shown in Table 1) [26,27,28], 12.5 µL of PCR Mix Plus (1.25 U) (A&A Biotechnology, Gdynia, Poland), and ultrapure water. For genetic material amplification, an iCycler (Biorad Laboratories Inc., Hercules, CA, USA) was used. The PCR products were electrophoresed in a 1.5% agarose gel stained with SYBR Green (Sigma Aldrich, Poznan, Poland). The PCR products were isolated from the agarose gels using a Gel Out Concentrator Kit (A&A Biotechnology, Gdynia, Poland) and were subsequently sent to Genomed (Warszawa, Poland) for sequencing with both forward and reverse PCR primers. The sequences obtained were compared to sequences from the GenBank database of the National Center for Biotechnology Information (NCBI) to confirm the probiotic species.

### 2.2. Experimental Design and Probiotic Embryo Intake Effectiveness Control Tests

The experiment was carried out on Ross 308 broiler breeder eggs in two replications, which were separate experiments. The eggs underwent incubation under large-scale commercial hatchery conditions (ModernHatch, Niemodlin, Poland) using HatchTech incubators with a computer system modulating the conditions for optimal temperature (37.8 °C or 100 °F) and humidity (55–60%) according to readings from sensors inside the machines. On the 18th day of incubation, the eggs were divided into four experimental groups, each comprising 20 000 eggs, and probiotics suspended in 0.9% mL of physiological saline (0.9% NaCl) were administered in ovo to the amniotic fluid. The eggs were injected with a 0.5 mL solution containing 3 × 10^6^ *Bacillus subtilis*, a 0.5 mL solution containing 3 × 10^6^ *Lactobacillus fermentum* (LAC group), a 0.5 mL solution containing 3 × 10^6^ *Enterococcus faecium* (ENT group), and 0.5 mL of 0.9% NaCl as a control group (C). The groups were labeled and incubated in HatchCare hatchers with immediate access to feed and water in an environment with fresh air and LED illumination. Twenty-four hours after in ovo probiotic administration, 10 embryos from each group were randomly selected and used for microbiological and PCR examination. After decapitation, the intestinal tract of each embryo was collected. Then, swabs were taken from a sterilely opened gizzard, small intestine, and cecum and used for microbiological culture. The samples were immediately diluted with peptone water, spread onto an MRS agar plate (Oxoid, Hampshire, UK) and BHI agar (Oxoid, Hampshire, UK) and then incubated for 48–72 h at 37 °C under aerobic microaerophilic (by using an anaerobic jar) conditions [29]. Additionally, genetic material was extracted from the samples of the gizzard, small intestine, and cecum of each embryo using the Genomic Mini Kit according to the manufacturer’s instructions, and PCR was performed with specific primers, as previously described (Table 1), to confirm the presence of probiotic genetic material.

### 2.3. Poultry Farming Conditions

After hatching, the chickens in each experimental group were placed in different poultry houses on commercial farms (each replication on a different farm belonging to one owner). The birds were reared under the same environmental conditions specified for the Ross 308 line; fed commercial diets (starter, grower, and finisher from the same supplier) according to the animal welfare recommendations of the European Union Directive 86/609/EEC; and provided with adequate husbandry conditions, with continuous monitoring of the stocking density, litter (straw), ventilation, etc. The birds were fed and given water ad libitum. The experiment lasted 35 days. The experiment was carried out with the consent of the Local Ethics Committee for Animal Experiments (Permission number 104/2017, Wrocław, Poland: approved on 25 October 2017).

### 2.4. Materials for Histological Studies

Eight chickens from each group were randomly selected and subjected to necropsy on the 7th (D7), 21st (D21), and 35th (D35) days after hatching. Sections of the thymus, bursa of Fabricius, cecal tonsil (CT), and spleen, as well as ca. 2 cm long sections of the duodenum, jejunum, and ileum, were taken for histological examination. Samples of the small intestine were taken from the midpoint of the duodenum, the midpoint of the jejunum between the point of entry of the bile duct and Meckel’s diverticulum, and the midpoint of the ileum between Meckel’s diverticulum and the ileocecal junction. The organ samples were fixed in 4% neutral-buffered formaldehyde and embedded in paraffin. Sections (5 µm thick) of each tissue were H&E-stained with hematoxylin according to Mayer (Roth GmbH, Karlsruhe, Germany) and eosin (Poch S.A., Gliwice, Poland). The slices were examined and photographed under a Nikon Eclipse 80i light microscope (Nikon, Melville, NY, USA) with a video camera. For immunohistochemical staining, second samples of cecal tonsils and spleens were fixed in 4% phosphate-buffered paraformaldehyde (pH 7.4) for 1 h, washed in 0.1 M phosphate buffer, and infiltrated with buffered 30% sucrose. Then, they were frozen in a cryostat (Leica CM1850, Leica Microsystems GmbH, Wetzlar, Germany) and cut into 10 μm serial sections, air-dried overnight, and frozen.

### 2.5. Immunohistochemical Staining

First, the cryosections were brought to room temperature, followed by the quenching of endogenous peroxidase in a 3% hydrogen peroxide solution and the blocking of non-specific binding via pre-incubation with an Antibody Diluent with a Background Reducing Component (Agilent, Santa Clara, CA, USA) for 20 min. Then, the serial sections were covered with monoclonal mouse anti-chicken antibodies (Southern Biotech, Birmingham, AL, USA) directed against the antigens Bu-1 (chicken B-cell marker, also known as chB6) (clone AV20, 1:500), CD4 (clone CT-4, 1:200), and CD8α (clone CT-8, 1:200) or PBS as a control and incubated for 1 h at room temperature. The antibodies were visualized using an EnVision™ System (Agilent) with the 3,3′-diaminobenzidine (DAB) chromogen according to the manufacturer’s instructions.

### 2.6. Morphometry

In the thymus, the cortex/medulla diameter ratio was measured in 3 lobules from each of the eight samples. In the bursa of Fabricius, the ratio of the cortex/medulla area was measured in transverse sections of three individual follicles from each of the eight samples. These data were collected on microphotographs (magnification of 40× or 100×) using the NIS-Elements AR 2.30 (Nikon) imaging software. Immunohistochemically stained sections of CTs and spleens were also analyzed. The area occupied by the antigen-positive (brown-colored) cells was estimated within an area of 0.29 mm^2^ (200× magnification) using the NIS-Elements AR 2.30 program and expressed as a percentage of the field of view. The germinal center surface area in the spleen was calculated for an area of 1.16 mm^2^ (100× magnification) and also expressed as a percentage of the field of view. All artifacts were eliminated by a histologist. For CTs, the fields of view were always set starting from the lamina propria mucosae in the direction of the lumen of the organ. Measurements performed on the small intestine included the height and width of intestinal villi, intestinal crypt depth, and goblet cell number. The length of the villi was estimated in a cross-section of the intestine using six randomly selected villi per bird. Similarly, the width of the villi was measured at half of their length (Appendix A). Based on these data, the average villus surface area (VSA) was calculated according to the formula given by [30]: VSA = (2π) × (VW/2) × (VH), where VW = villus width, and VH = villus height. The depth of intestinal crypts was measured from the base upward to the region of transition between the crypt and villus [31]. The density of goblet cells was calculated per 1 linear mm of epithelium that covered the intestinal villi.

### 2.7. Isolation and Identification of Probiotic and Campylobacter Strains from the Intestinal Tract

On D7, D21, and D35 after hatching, 30 chickens from each group were randomly selected for cloacal swab collection and used for PCR. The experiment was performed in duplicate. The samples were cultured in BHI broth for 24 h and then streaked onto an MRS agar plate and BHI agar for culture and antimicrobial susceptibility detection, as described previously, to confirm the presence of the administered probiotic strains. For the quantitative isolation of *Campylobacter* spp., 1 g of cecal content was homogenized in 9 mL of sterile 0.9% NaCl to create a uniform suspension, following the ISO 10272-2:2017 standard of International Organization for Standardization. Tenfold serial dilutions of this suspension were prepared, and 0.1 mL was plated onto mCCDA Agar Base (Oxoid, Hampshire, UK) containing a *Campylobacter*-selective supplement comprising cefoperazone and amphotericin B (CCDA-selective supplement SR0155E). The inoculated plates were incubated under microaerophilic conditions at 42 °C for 48 h. After incubation, *Campylobacter* colonies were counted, and the colony-forming units per gram (CFU/g) of feces were calculated to quantify the bacterial load. The molecular identification of the *Campylobacter* genus and species was made using specific primers (Table 2) [32,33]. PCR reactions were performed using an iCycler in a 25 µL reaction mixture containing 5 ng of template DNA, 10 pmol of each primer, 12.5 µL of PCR Mix Plus (1.25 U) (A&A Biotechnology, Gdynia, Poland), and ultrapure water. The *Campylobacter* genus PCR conditions were an initial DNA denaturing step at 95 °C for 5 min, followed by a 33-cycle reaction (94 °C for 1 min, 58 °C for 1 min, 72 °C for 2 min), with the final extension step at 72 °C for 2 min. The *C. jejuni* and *C. coli* PCR conditions were initial denaturation at 95 °C for 6 min, followed by 30 cycles of annealing (95 °C for 30 s, 59 °C for 30 s, and 72 °C for 30 s), with final extension at 72 °C for 7 min. The strains used as positive controls were *C. jejuni* ATCC 33560 and *C. coli* ATCC 33559. The PCR products were electrophoresed in a 1.5% agarose gel stained with SYBR Green.

### 2.8. Statistical Analysis

The data from morphometric measurements were analyzed using Statistica 13.1 software (StatSoft Polska Sp. z o.o., Cracow, Poland). The Brown–Forsyth test was used to determine the homogeneity of variance between groups. Multiple comparisons between groups were assessed using one-way ANOVA with post hoc Tukey tests for data that met the criteria of a normal distribution and homogeneity of variance or the Kruskal–Wallis test for ranks (one-way ANOVA on ranks) for data that did not meet these assumptions. A value of *p* < 0.05 was considered significant. The villus-to-crypt ratio was calculated for each individual as the ratio of the mean villus height to the mean crypt depth from 6 measurements. Only these individual results were used to calculate the mean for the whole group.

## 3. Results

### 3.1. Probiotics

The PCR confirmed that the probiotics belonged to the correct species according to the order. The probiotic strain PCR products showed 100% homology, with fragments of *Bacillus subtilis* isolates (OP027905.1, KT763078.1), *Enterococcus faecium* (CP131117.1, CP131093.1), and *Lactobacillus fermentum* (CP021104.1; LT906621.1).

### 3.2. Experimental Design and Probiotic Embryo Intake Effectiveness Control Tests

The PCR testing of gastrointestinal content revealed the presence of probiotic genetic material in nine embryos (9/10, 90%) for *Bacillus subtilis* and in all embryos (10/10, 100%) for *Lactobacillus fermentum* and *Enterococcus faecium*. In the group that received *B. subtilis*, 60% (6/10) of the embryos took up the probiotic in an amount that allowed its detection in microbiological culture. In the two groups that received *L. fermentum* and *E. faecium*, 90% (9/10) of the embryos took up the probiotic in an amount that allowed its detection in microbiological culture.

### 3.3. Histology

Morphological evaluation of the thymus, bursa of Fabricius, cecal tonsil, spleen, duodenum, jejunum, and ileum was performed using H&E staining. No pathological changes were detected in the histological examination of these tissues from any group at any of the time points studied.

### 3.4. Morphometry of the Intestines

In all regions of the small intestine, several parameters were analyzed, including the villus surface area, crypt depth, villus-to-crypt ratio, and goblet cell number. On D7, a decrease in crypt depth in the jejunum and an increase in the number of goblet cells in the duodenum were noted in the BAC group compared to the C group (Figure 1 and Figure 2). On D21, similar effects were present in both the BAC and LAC groups, where an increase in villus surface area in the ileum and a decrease in goblet cell number in the jejunum were observed. At the same time, in the ENT group, a decrease in villus surface area in the duodenum and an increase in the crypt depth in the ileum were observed. The most visible effects of in ovo probiotic administration on the small intestine were observed on D35. In the BAC group, a reduction in crypt depth in the duodenum was found, which resulted in an increase in the villus-to-crypt ratio. In the LAC group, the villus surface area in the duodenum and ileum significantly increased, as did the goblet cell number in the ileum. In the ENT group, the villus surface area increased in the duodenum and jejunum, as did the villus-to-crypt ratio in the jejunum.

### 3.5. Morphometry of Immune Organs

Statistical analysis of the cortex/medulla diameter ratio in thymus lobules revealed no differences between groups at any time point (Table 3). In bursal follicles, the cortex/medulla area ratio was similar in all groups studied, except for D7, when the ratio was lower in the ENT than in the C group (Table 3). In the cecal tonsil, the number of B cells (Bu1^+^) was significantly higher in the ENT than in the C group on D35 after hatching (Figure 3). The share of Th cells (CD4^+^) was higher in the BAC group on D7, while the cytotoxic cell levels (CD8α^+^) were similar in all groups and at all time points studied. In the spleen, the number of B cells (Bu1^+^) was significantly higher in the BAC group than in the C group on D21 and D35 and in the LAC group on D35 (Figure 3). The number of Th lymphocytes decreased in the BAC group on D21, while the cytotoxic cells (Tc and NK cells) from probiotic-treated groups did not differ from those in the C group. Similarly, the mean area of spleen germinal centers in the field of view did not differ between groups on D21 and D35 (Table 3). On D7, the germinal centers were not visible yet.

### 3.6. Isolation and Identification of the Culture of Campylobacter spp. from the Intestinal Tract

On D7, *Campylobacter* spp. were not microbiologically detected in either the C or BAC group (Figure 4). In the LAC group, only two samples were positive (2/30, 6.67%), with the level of *C. jejuni* ranging from 2 to 3.3 log CFU/g, while in the ENT group, three samples (3/30, 10%) exhibited levels of *C. jejuni* between 2 and 3.1 log CFU/g. On D21, no *Campylobacter* spp. were observed in the LAC group, while in the C group, *C. jejuni* was present in three samples (10%) at levels ranging from 2 to 3 log CFU/g. In the BAC group, *C. jejuni* was present only in one sample (3.33%) at a level of 2 log CFU/g. The highest number of positive samples, as many as nine (9/30, 30%), was found in the ENT group, with *C. jejuni* levels ranging from 2 to 3.1 log CFU/g. Isolation studies on D35 demonstrated that the lowest level of cecal colonization by *C. jejuni* was in the control group, where it occurred in only two samples (6.67%) and ranged from 2 to 2.8 log CFU/g. *Campylobacter* colonization in the cecum was the highest in the BAC group, with eight samples (26.67%) showing colonization at levels ranging from 5 to more than 6 log CFU/g. The LAC and ENT groups exhibited significantly higher (*p* < 0.01) *Campylobacter* growth compared to the C group. In the ENT group, *Campylobacter* spp. were identified in 21 samples (21/30, 70%), with colonization levels ranging from 3 to 5 log CFU/g in cecal contents. In the LAC group, the level of colonization of the cecum by *Campylobacter* ranged from over 6 log CFU/g, with as many as 21 positive samples (70%). PCR confirmed all positive *Campylobacter* spp. samples as *Campylobacter jejuni.* The highest percentage of positive samples for probiotic presence was in the ENT and LAC groups. In the ENT group, 70% of samples on D7, 87% of samples on D21, and 97% of samples on D35 were positive for the presence of the *E. faecium* probiotic. In the LAC group, 83% of samples on D7 and D21 and 100% on D35 were positive for the *L. fermentum* probiotic. The BAC group showed the lowest detection level of the probiotic (30% on D7, 40% on D21, and 80% on D35).

### 3.7. Hatchability and Final Production Parameters

In ovo probiotic administration did not affect embryo development. The data from the hatchery revealed that hatchability in all in ovo -inoculated groups (89.7%) did not significantly differ (*p* > 0.05) from that in eggs not subjected to the in ovo procedure (89.9%). In the first trial, the BAC group exhibited a higher average slaughter weight (2.60 kg) but also a higher mortality rate (4.14%) compared to the control group, where the average slaughter weight was 2.52 kg, and the mortality rate was 3.72% (Table 4). The European Broiler Index (EBI) in the BAC group was also higher (412.50) than that in the control group (404.09). In the second trial, better production results were observed in the LAC group, where the mortality rate was 2.6%, compared to 2.8% in the control group. The EBI in the LAC group was slightly higher (388.41) than that in the control group (384.11).

## 4. Discussion

Probiotics can be delivered to the host in the posthatch period with feed or water or as early as the stage of embryonic development using the in ovo technique. Inoculation with bioactive compounds on day 12 of embryonic development is referred to as in ovo stimulation, while a similar procedure performed on days 17–18 is called in ovo feeding [34,35]. It is postulated that probiotics present in the intestinal lumen can affect the host through the following mechanisms: the competitive exclusion of pathogenic bacteria, competition for growth factors and nutrients, the production of antimicrobial substances, the enhancement of adhesion to the intestinal mucosa, improvement in epithelial barrier function, and improvement in the secretion of IgA [36,37]. Probiotics have also been shown to exert systemic effects by modulating the immune system in the form of an increased antibody response, decreased inflammation, and the stimulation of phagocytosis [36,38]. This strengthening of the immune system’s potential enables a more effective response to pathogenic bacteria. For example, in a study conducted by de Oliveira et al. [39], it was demonstrated that in ovo delivery of probiotics (*Enterococcus faecium* and *Bacillus subtilis*) significantly reduced mortality when challenged with *Salmonella* on day 4 posthatch. The maturation of the gastrointestinal tract in chickens depends strongly on exogenous feed intake, which sometimes takes place in commercial production even 72 h after hatching [40]. The morphological manifestation of these changes is an increase in the length of the villi and an increase in the depth of crypts with an accompanying increase in enzymatic activity in the small intestine [41,42]. Moreover, it was previously shown that bacterial colonization of the intestine increases the villous height by stimulating proliferation, increasing the migration rate, and prolonging the lifespan of enterocytes [43,44]. It was also found that the chick embryo contains microbiota with at least 30 different phylotypes, so it does not develop, as was originally thought, in sterile conditions [45]. Therefore, the use of the in ovo feeding method enables the shaping of the intestinal microbiota even before hatching. This study indicated that in ovo inoculation with *L. fermentum* significantly increased the villus surface area in the jejunum on D21 and in the ileum on D21 and D35. The presence of this bacterium also resulted in an increase in the goblet cell density in the epithelium that covers the intestinal villi. This effect is beneficial because an increase in the surface area of the mucosa significantly facilitates the absorption of nutrients, which occurs primarily in the small intestine. In addition, the increased number of goblet cells enables the production and release of an appropriate amount of mucus to cover the surfaces of enterocytes. This layer of mucus protects the mucosa from mechanical damage and dehydration and protects the underlying epithelium from luminal contents, which include pathogenic bacteria, viruses, and parasites [46,47]. Enabling the efficient absorption of nutrients, vitamins, and minerals, as well as ensuring an appropriate barrier function, is invaluable in broilers, where achieving the intended production parameters and slaughter traits, along with maintaining proper health conditions, is closely related to the efficient functioning of the intestine. Other authors also reported the positive effect of in ovo probiotic administration on small intestine morphology [48]. These studies indicated that the application of *L. plantarum* combined with *L. salivarius* (10^9^ cfu) resulted in an increase in villus height in the jejunum of broiler chickens on the 35th day after hatching. Feed supplementation with *Lactobacillus sp.* also had beneficial effects on broilers, resulting in an increase in the villus height/crypt depth ratio and villus height in both the duodenum and ileum [49]. Previous studies in rats also revealed that the oral administration of a probiotic mixture enhances the integrity of the intestinal epithelial barrier by stimulating mucus production in goblet cells [50]. Similar results obtained by Ariyadi and Harimurti [51] showed that the addition of indigenous probiotic lactic acid bacteria to the feed increased the villus height and width in the duodenum, jejunum, and ileum and improved the expression of mucin mRNA in the ileum. Based on our results and the literature data, it can be concluded that *Lactobacillus* bacteria increase both the number of goblet cells and the amount of mucus produced by individual goblet cells in the intestine. This is particularly important if we consider the protective role of mucus that covers the intestinal epithelium. This layer prevents the direct contact of pathogenic bacteria with the surfaces of enterocytes and therefore limits their adhesion to these cells, which is the first step of invasion [52]. The influence of *E. faecium* supplementation on the villus surface area was not as clear as in the case of the previously discussed probiotic bacteria. Specifically, on D21, the surface area of villi decreased in the duodenum, while on the last day of the experiment, it significantly increased in the duodenum and jejunum, with an accompanying increase in the villus-to-crypt ratio in the jejunum. Thus, for this probiotic, the beneficial increase in the absorption surface of the small intestine was achieved later than in the case of *L. fermentum* supplementation. Ambiguous effects of probiotic administration on intestinal morphology were noted after inoculation with *B. subtilis*. Favorable increases in the villus surface area in the ileum on D21 and in goblet cell number in the duodenum on D7 were accompanied by decreases in crypt depth in the jejunum on D7 and in the duodenum on D35 and a severe reduction in the number of goblet cells in the jejunum on D21, which calls into question the preventive use of this probiotic to improve intestinal structure and function. In conclusion, the in ovo administration of probiotics on the 18th day of incubation had a beneficial effect on the development of the small intestine in broilers hatched under early feeding conditions, but this effect was not clearly visible before 21–35 days after hatching. In the first period of life, contact with food probably has the greatest impact on intestinal development, while the process of determining the composition of the intestinal microbiota and its impact on the structure of the mucous membrane requires several weeks. The presented results are also consistent with the concept of “competitive exclusion”, which asserts the protective effect of commensal bacteria against pathogenic strains such as *Salmonella* [53,54]. Previous studies by Bielke [55] suggested that providing competitive exclusion cultures to chicks increases protection only when given prior to exposure to pathogenic bacteria. In this context, the use of the in ovo technique, which guarantees the supply of probiotic bacteria to the chicken embryo in the period before hatching, seems to be a very good choice. Several studies demonstrated that probiotic administration affects the structure and function of the immune system of poultry. Awad et al. [49] reported that oral *Lactobacillus* sp. supplementation slightly improved the absolute and relative weights of the thymus and spleen in broiler chicks. In ovo administration of lactobacilli to chick embryos might be beneficial for accelerating the development and immunological maturation of the bursa of Fabricius. It was proven that *L. acidophilus* and a mixture of *Lactobacillus* species elicited higher expression of genes responsible for B-cell development, differentiation, and survival (B-cell activating factor (BAFF), as well as BAFF receptor (BAFF-R)) and antibody production (IL-10) and diversification (TGF-β) [56]. Increased expression was also noted in T helper cell-associated cytokine genes, including IFN-γ, IL-12p40, IL-4, IL-13, and IL-17 [56]. Another experiment revealed that in ovo administration of a lactobacilli cocktail on the 18th embryonic day increased the expression of IL-2 in the bursa of Fabricius [57]. In this study, the cortex/medulla area ratio in bursal follicles was similar in all groups studied, except on D7, when the ratio was lower in the ENT than in the HC group. In the first three weeks of life, with the development of the bursa of Fabricius, the surface of the cortex and the cortex/medulla ratio increase, which is related to stimulation with antigens present in food and the environment [23]. The administration of prebiotics may also indirectly affect the bursa of Fabricius by shaping the intestinal microbiota. Therefore, the observed reduction in the cortex/medulla ratio in the ENT group may indicate the slower development of this organ resulting from reduced stimulation by food antigens. Another explanation for the reduction in cortical thickness may be related to the migration of B lymphocytes within the follicles, which involves these cells first passing from the medulla to the cortex and only then being released into the peripheral blood. The decreased ratio in the ENT group may therefore result from faster emigration of mature B cells from the bursa to secondary (peripheral) lymphoid organs at this time point. Alizadeh et al. [57] found that in ovo administration of a lactobacilli cocktail on the 18th embryonic day resulted in the downregulated expression of several immune-related genes in the spleen, such as interferon beta (IFN-β), interleukin (IL) 18, and transforming growth factor beta (TGF-β). Moreover, in the spleen, a higher frequency and absolute number of macrophages were noted, but the frequency and absolute numbers of Bu-1^+^IgM^+^ B cells, CD3^+^CD8^+^ T cells, and CD3^−^CD8^+^ T cells were not affected in the treatment groups [57]. Probiotics can also modulate the number of immune cells in the gastrointestinal tract. Bai et al. [58] reported that the dietary inclusion of *Lactobacillus fermentum* and *Saccharomyces cerevisiae* increased the proportions of CD3^+^, CD4^+^, and CD8^+^ T lymphocytes in the intestine of broilers on days 21 and 42 after hatching. Studies in broilers indicated that the in ovo distribution of the preparation Primalac W/S, containing a mixture of *Lactobacillus acidophilus*, *Lactobacillus casei*, *Enterococcus faecium*, and *Bifidobacterium bifidum*, modulated immune-related gene expression in the cecal tonsils. Specifically, the downregulated expression of Toll-like receptors-2 and -4, inducible nitric oxide synthase (iNOS), trefoil factor-2, mucin-2, IFN-*γ*, IL-4, and IL-13 was noted [59]. Previous studies in broilers revealed that probiotics (*Lactococcus lactis subsp. cremoris*) delivered in ovo on day 12 of egg incubation increased the number of CD4^+^ cells in the CT on day 7 after hatching [60]. However, there were no changes in the number of Bu-1^+^, CD4^+^, or CD8^+^ cells in the CT on days 21 and 42 or in the spleen on days 7, 21, and 42 after hatching. The probiotic also had no influence on the germinal center surface area in the spleen. A similar experiment in Green-legged Partridge-like chicks demonstrated that probiotic supplementation did not affect the number of cells in the CT and spleen or the area of germinal centers in the spleen. The present experiments indicated that the selected probiotics may influence the number of B cells in peripheral immune organs, mainly on D35. Increases in B-cell numbers were found at this time point in the cecal tonsil in the BAC and ENT groups and in the spleen in the BAC and LAC groups. We previously conducted an experiment investigating the influence of early posthatch feeding technology on the immune system of broilers [23]. Interestingly, increases in the weight of the bursa and CD4^+^ cell number in the CT were noted on day 7 only, suggesting the more rapid development of these organs due to early stimulation of the gastrointestinal tract by contact with feed. The current experiments indicated that *B. subtilis* supplementation enhances this effect by further increasing the colonization of the CT in CD4^+^ cells. Therefore, it can be concluded that the administration of probiotics in ovo combined with early posthatch feeding has a positive effect on the development of the immune system. In the spleen, these changes were only evident in the second half of fattening, whereas in the gut-associated lymphoid tissue (GALT), they were noted both at the beginning and at the end of this period. It is likely that probiotic administration induces subtle changes in the rate of colonization of peripheral lymphoid organs, which usually only becomes apparent several weeks after hatching, especially in organs such as the spleen that do not have direct contact with the gut microbiota. It can therefore be assumed that these changes are at least partially time-dependent. A variety of bacteria (*Bacillus* spp., *Bifidobacterium* spp., *Enterococcus* spp., *Lactobacillus* spp., *Streptococcus* spp., and *Lactococcus* spp.) have been tested as probiotics in poultry [61]. In this study, administering the selected probiotics (*B. subtilis, L. fermentum*, and *E. faecium*) in ovo on the 18th day of incubation did not reduce the colonization of the birds’ gastrointestinal tracts by *Campylobacter* spp. on D35, at the end of rearing. On day 35, higher colonization rates and bacterial loads were observed across all experimental groups, with the percentages of positive samples reaching 26.67%, 70%, and 70% in BAC, LAC, and ENT, respectively. In contrast, the control group showed only 6.67% positive samples. However, in the LAC group, on D7, *C. jejuni* growth was observed in only two samples, with levels ranging from 2 to 3.3 log_10_ CFU/g. On D21, no *C. jejuni* growth was detected in any of the examined samples in the LAC group. This may indicate alternating inhibitory effects on the colonization and proliferation of *Campylobacter* spp. following the in ovo administration of *Lactobacillus fermentum* on the 18th day of incubation. The results of our study may suggest that certain probiotic strains could inadvertently enhance *C. jejuni* colonization under specific conditions, such as the type of probiotic strain used, its ability to adhere to the intestinal mucosa, the route of administration, the dosage, and the timing. For instance, Robyn et al. [62] observed that a live *Enterococcus faecalis* strain did not reduce *C. jejuni* colonization in broilers and, in some cases, may have even facilitated it. Additionally, Ganan et al. [63] reported that simultaneous exposure of probiotics and *C. jejuni* to intestinal mucus increased the pathogen’s adhesion. The authors suggested that this might be due to the co-aggregation of the probiotics and *C. jejuni*, resulting in bacterial clusters that adhere more readily to the mucus surface. Interesting results were reported by Olsen et al. [64], who found that the daily administration of the probiotic strain *Enterococcus faecium* 669 in drinking water significantly reduced the colonization and shedding of *Salmonella Enteritidis* in broiler chickens. This suggests that continuous probiotic supplementation can effectively lower the pathogen load in poultry. Moreover, the authors demonstrated that probiotic treatment improved intestinal barrier integrity, as evidenced by the increased expression of tight junction- and mucosal barrier-related genes (e.g., *mucin-2*, *zonula occludens-1*, and *claudin-5*) and favorable changes in jejunal villus morphology. Recent research on the impact of probiotics on gut health was published by Lyte et al. [65], who demonstrated that a dopamine-producing *Enterococcus faecium* probiotic can effectively convert L-dopa into dopamine in the gut, significantly reducing inflammation-associated norepinephrine levels induced by a high non-starch polysaccharide diet. This represents the first application of a microbial endocrinology-based probiotic to modulate the neurochemical pathways involved in gut inflammation in poultry. On the contrary, many researchers report the beneficial role of probiotics in limiting *Campylobacter* spp. shedding, suggesting them as a tool for effectively combating the potential contamination of poultry products [66,67,68,69]. The studies conducted by Śmiałek et al. [68] demonstrated that adding a multispecies probiotic composed of *Lactococcus lactis*, *Carnobacterium divergens*, *Lactobacillus casei*, *Lactobacillus plantarum*, and *Saccharomyces cerevisae* to broiler feed was able to reduce the extent of *Campylobacter* spp. invasion in the gastrointestinal tract. Baffoni et al. [66] recognized the benefit of early-life synbiotic administration (*Bifidobacterium* strain combined with Xylo-oligosaccharides (XOS) or Galacto-oligosaccharides (GOS)), which increased the abundance of beneficial bacteria (i.e., *Bifidobacterium* spp. and *Lactobacillus* spp.) that competitively reduced *C. jejuni*. Some authors have highlighted the advantages of using several selected probiotics simultaneously. Neveling and Dicks [70] revealed that a combination of *Enterococcus faecium*, *Pediococcus acidilactici*, *Bacillus animalis*, *Lactobacillus salivarius*, and *Lactobacillus reuteri* reduced colonization by *Campylobacter jejuni* and *Salmonella* Enteritidis in the gastrointestinal tract of broilers. Meanwhile, *Bacillus subtilis* was found to enhance feed conversion, improve intestinal morphology, boost the immune response, and inhibit colonization by *Campylobacter jejuni, Escherichia coli*, and *Salmonella* Minnesota. The differences between the results obtained in our research and those by other authors may stem from various factors, including the type of substance used (probiotics, prebiotics, synbiotics), as well as the timing and method of administration, which can be crucial for probiotic availability and effectiveness, including in reducing intestinal colonization by foodborne bacteria such as *Campylobacter* spp. In our study, probiotics were administered in ovo to promote early gut colonization in birds, while most other experiments used the oral route—via feed or gavage. Our conclusions appear to be supported by Kpodo and Proszkowiec-Weglarz [71], who stated that the in ovo delivery of probiotics has the potential to reduce intestinal pathogens such as *Salmonella* and *E. coli*. However, the outcomes depend on several factors, including the specific strains used, timing, dosage, and interactions with host physiology, resulting in effects that may be promising, limited, or even adverse in terms of pathogen exclusion. Overall, while probiotics offer a valuable tool for promoting optimal growth and disease resistance in poultry, their application must be carefully tailored. Effective use requires thoughtful strain selection, appropriate delivery methods, and comprehensive evaluation to balance benefits with potential risks. In our study, in ovo probiotic supplementation did not influence hatchability, but *B. subtilis* supplementation contributed to a higher average slaughter weight, while in ovo administration of *L. fermentum* improved bird survival during rearing. Similarly, Pender et al. [59] found that the in ovo administration of probiotics did not impact hatchability but led to improved body weight gain during the first week posthatch. Their study also revealed the modulation of immune-related gene expression in the ileum and cecal tonsils, suggesting that in ovo probiotic supplementation can influence early immune development. Additionally, Muyyaikkandy et al. [72] found that in ovo probiotic supplementation was associated with enhanced muscle development in broiler embryos. Early feeding also has a positive impact on muscle development, which was confirmed by Gaweł et al. [25]. The authors reported that administering probiotics in ovo promoted embryonic muscle growth, potentially leading to improved meat production posthatch.

## 5. Conclusions

This study revealed that in ovo probiotic administration on the 18th day of incubation, combined with early posthatch feeding, had a beneficial effect on the development of the small intestine and peripheral immune organs. In the spleen, an increased number of B cells was noted, while in the cecal tonsils, a higher number of B and CD4^+^ cells was found. The probiotics used did not reduce the colonization of the intestinal tract by *Campylobacter* spp. Moreover, the in ovo procedure did not adversely influence the hatchability of the broilers.

## Figures and Tables

**Figure 1 microorganisms-13-01219-f001:**
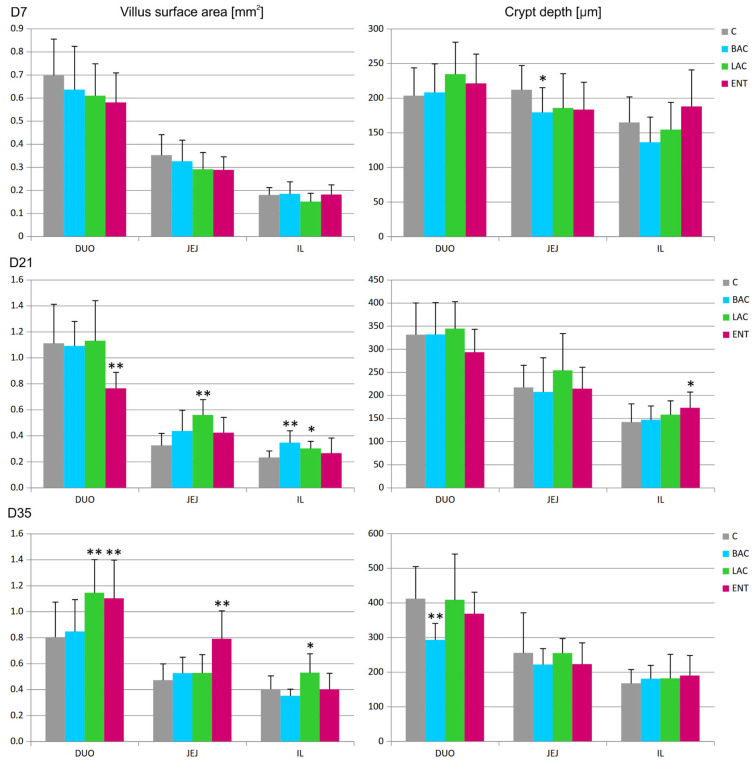
Effect of early feeding (HatchCare) method combined with probiotics on average villus surface area and average crypt depth in duodenum (DUO), jejunum (JEJ), and ileum (IL). C—control; BAC—*Bacillus subtilis*; LAC—*Lactobacillus fermentum*; ENT—*Enterococcus faecium* group. D7-D35—days 7–35. Significant difference compared to C group (* *p* < 0.05; ** *p* < 0.01), n = 8 for each group and time point.

**Figure 2 microorganisms-13-01219-f002:**
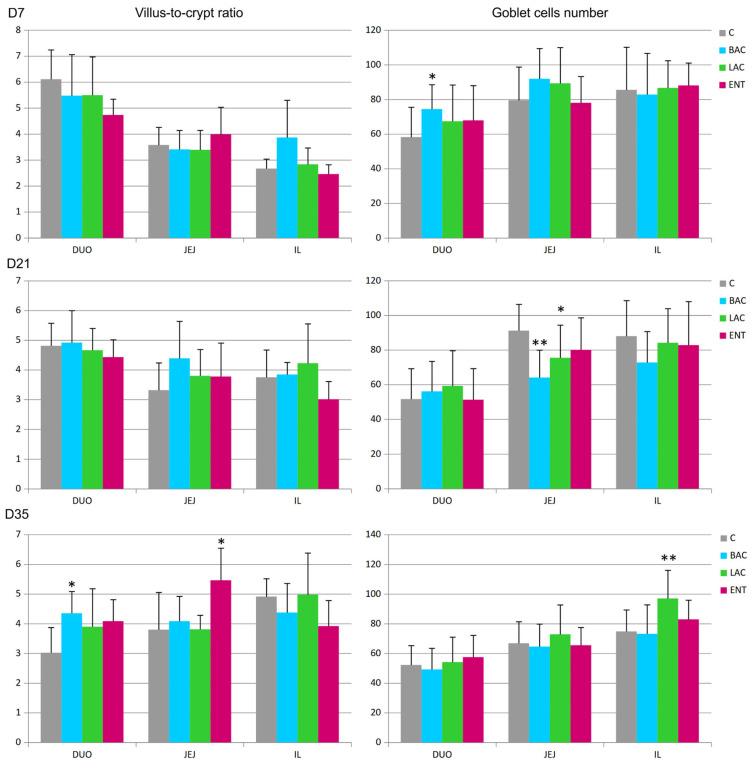
Effect of early feeding (HatchCare) method combined with probiotics on villus-to-crypt ratios and goblet cell numbers in duodenum (DUO), jejunum (JEJ), and ileum (IL). C—control; BAC—*Bacillus subtilis*; LAC—*Lactobacillus fermentum*; ENT—*Enterococcus faecium* group. D7-D35—days 7–35. Significant difference compared to C group (* *p* < 0.05; ** *p* < 0.01), n = 8 for each group and time point.

**Figure 3 microorganisms-13-01219-f003:**
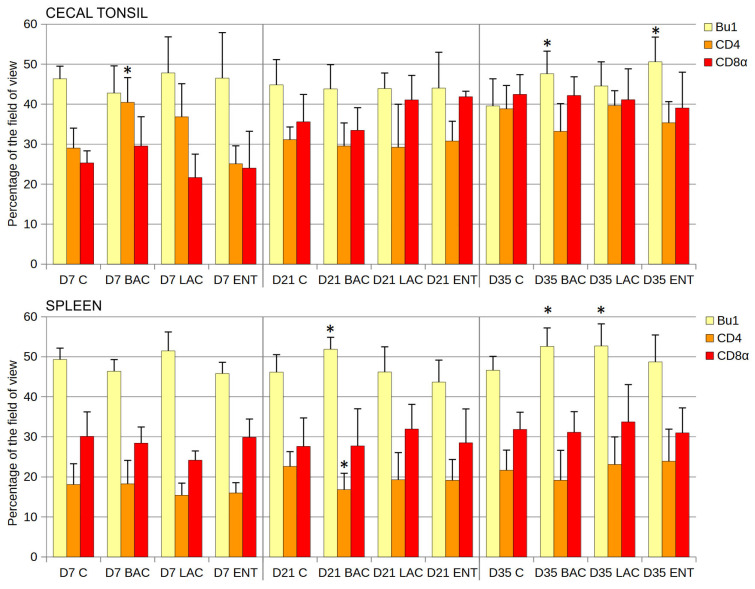
Effect of early feeding (HatchCare) method combined with probiotics on immune cell composition in chicken cecal tonsil and spleen. Area occupied by Bu-1-, CD4-, and CD8α-positive cells (mean ± SD, n = 8) on the 7th day (D7), 21st day (D21), and 35th day (D35) after hatching in control (C), *Bacillus subtilis* (BAC), *Lactobacillus fermentum* (LAC), and *Enterococcus faecium* (ENT) groups. Significant difference compared to C group (* *p* < 0.05), n = 8 for each group and time point.

**Figure 4 microorganisms-13-01219-f004:**
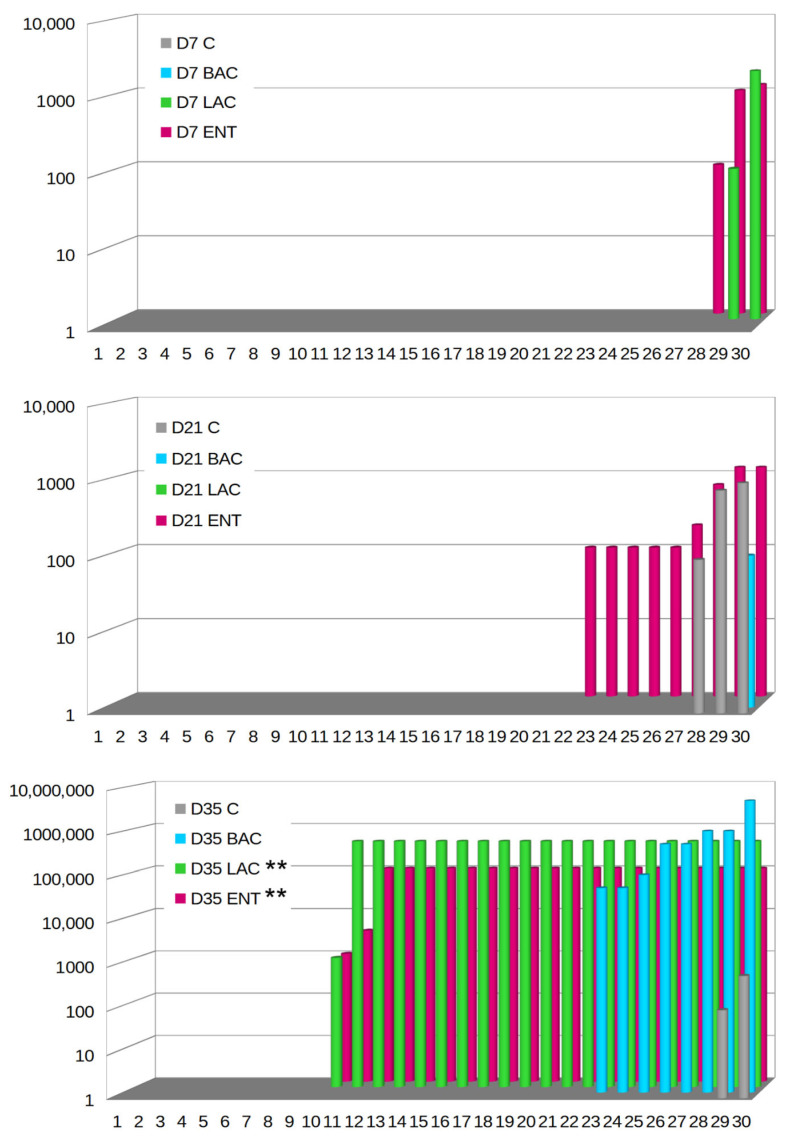
*Campylobacter jejuni* colony counts in 30 broilers (arranged in ascending order) after in ovo administration of selected probiotics. D7-D35—days 7–35. C—control; BAC—*Bacillus subtilis*; LAC—*Lactobacillus fermentum*; ENT—*Enterococcus faecium* group. Significant difference compared to C group (** *p* < 0.01).

**Table 1 microorganisms-13-01219-t001:** Species-specific primers of probiotics used in this study.

Bacterium Species	Primer Sequence (5′→3′)	Product Size (bp)	Reference
*Bacillus subtilis*	F:AAGTCGAGCGGACAGATGG R:CCAGTTTCCAATGACCCTCCCC	600	[27]
*Enterococcus faecium*	F:TTGAGGCAGACCAGATTGACG R:TATGACAGCGACTCCGATTCC	658	[26]
*Lactobacillus fermentum*	F:GACCAGCGCACCAAGTGATA R:AGCGTAGCGTTCGTGGTAAT	129	[28]

**Table 2 microorganisms-13-01219-t002:** *CCampylobacter* genus- and species-specific primers used in PCR.

Species	Primer Sequence (5′→3′)	Product Size (bp)	Gene	Reference
*Campylobacter* spp.	F:AAGTCGAGCGGACAGATGG R:CCAGTTTCCAATGACCCTCCCC	439	*16S rRNA*	[32]
*Campylobacter jejuni*	F:ACTTCTTTATTGCTTGCTGC R:GCCACAACAAGTAAAGAAGC	323	*hipO*	[33]
*Campylobacter coli*	F:GTAAAACCAAAGCTTATCGTG R:TCCAGCAATGTGTGCAATG	126	*glyA*	[33]

**Table 3 microorganisms-13-01219-t003:** Morphometric parameters for thymus, bursa of Fabricius, and spleen (mean ± SD, n = 8) of control (C), *Bacillus subtilis* (BAC), *Lactobacillus fermentum* (LAC), and *Enterococcus faecium* (ENT) groups. D7-D35—days 7–35 (mean ± SD, n = 8). Significant difference compared to C (* *p* < 0.05).

	D7	D21	D35
Cortex/medulla diameter ratio in thymic lobules			
C	0.95 ± 0.23	0.87 ± 0.32	0.89 ± 0.26
BAC	0.98 ± 0.26	0.88 ± 0.23	0.87 ± 0.26
LAC	0.96 ± 0.25	0.88 ± 0.22	0.88 ± 0.30
ENT	0.97 ± 0.28	0.92 ± 0.34	0.84 ± 0.24
Cortex/medulla area ratio in bursal follicles			
C	0.50 ± 0.13	0.53 ± 0.13	0.57 ± 0.19
BAC	0.53 ± 0.24	0.54 ± 0.19	0.55 ± 0.17
LAC	0.46 ± 0.17	0.56 ± 0.13	0.56 ± 0.19
ENT	0.40 ± 0.11 *	0.53 ± 0.11	0.62 ± 0.21
Germinal centers area in spleen (percentage of the field of view)			
C	(not visible)	2.63 ± 1.77	3.49 ± 1.35
BAC	—	2.19 ± 1.24	2.51 ± 1.43
LAC	—	2.34 ± 1.36	2.95 ± 1.94
ENT	—	2.25 ± 1.67	3.23 ± 1.63

**Table 4 microorganisms-13-01219-t004:** Final production parameters of control (C), *Bacillus subtilis* (BAC), *Lactobacillus fermentum* (LAC), and *Enterococcus faecium* (ENT) groups.

	Feed Conversion Ratio (FCR)	Mortality in the Flock (%)	Average Slaughter Weight (kg)	Fattening Days	European Broiler Index (EBI)
I repetition					
C	1.58	3.72	2.52	38	404.09
BAC	1.59	4.14	2.60	38	412.50
LAC	1.58	4.71	2.54	38	403.12
ENT	1.60	4.75	2.50	38	391.65
II repetition					
C	1.66	2.80	3.28	50	384.11
BAC	1.71	2.90	3.14	50	356.60
LAC	1.64	2.60	3.27	50	388.41
ENT	1.66	4.20	3.23	50	372.81

## Data Availability

The datasets generated during the current study are available from the corresponding author on reasonable request.

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
