# Peer review of "Boosting Broiler Health and Productivity: The Impact of in ovo Probiotics and Early Posthatch Feeding with Bacillus subtilis, Lactobacillus fermentum, and Enterococcus faecium"

_microorganisms, 2025, doi:10.3390/microorganisms13061219_

Round 1

Reviewer 1 Report

Comments and Suggestions for Authors

The manuscript by Madej et al. investigates the impact of in ovo administration of probiotics (Bacillus subtilis, Lactobacillus fermentum, and Enterococcus faecium) combined with early posthatch feeding on the health and productivity of broiler chickens. The study found that this treatment has a positive effect on the development of the small intestine and peripheral immune organs, but it did not reduce the colonization of Campylobacter spp.in the gut. The research also explores the effects of probiotics on broiler production performance, including hatchability, slaughter weight, and mortality. This is a meaningful study, but the manuscript as a whole has several shortcomings.

1 The introduction mentions that the traditional reliance on antibiotic growth promoters (AGPs) to enhance the growth performance of broiler chickens and prevent diseases is controversial, as it may lead to the emergence of antibiotic-resistant bacteria and residues in meat products. Are there any specific data or references to illustrate the severity of this controversy?

2 The introduction of the manuscript mentions that probiotics, as an alternative strategy to AGPs, have the potential to improve gut health, enhance immune responses, and improve growth performance. Could the authors please provide specific probiotic strains or research cases to support this view?

3 The introduction of the manuscript mentions that the in ovo administration of probiotics can establish early colonization of the gut microbiota, thereby providing health benefits from the onset of life. Is the application of this method in actual production limited? It is suggested to supplement the practical difficulties in the in ovo administration of probiotics and the problems that may be encountered, such as the survival rate of probiotics and their uniform distribution.

4 The introduction of the manuscript mentions that this study aims to assess the impact of in ovo probiotics administration on production parameters, intestinal condition, and the immune system in broiler chickens. Please clarify in the text what the hypothesis of the study is. What are the specific objectives of the study? What are the expected outcomes of the study? What are the innovative aspects of the study?

5 What is the basis for the concentration of probiotics used in the manuscript, as mentioned in the "Materials and Methods" section: "The eggs were injected with 0.5 mL solution containing 3x10⁶ Bacillus subtilis,0.5 mL solution containing 3x10⁶ Lactobacillus fermentum (LAC group), 0.5 mL solution containing 3x10⁶ Enterococcus faecium (ENT group)"? Does this method cause harm to the embryo?

6 In the manuscript's section on materials and methods, it is mentioned that 24 hours after in ovo administration of probiotics,10 embryos were randomly selected for microbiological and PCR testing. Is this testing method sufficiently sensitive? Can it accurately reflect the intake of probiotics? Can it distinguish between different types of probiotics? Please provide corresponding explanations in the text.

7 In the statistical analysis,were normality and homogeneity of variance tests conducted? Were corrections made for multiple comparisons?

8 The presentation of the results section is very messy, and the color schemes of the figures and tables are unattractive. Please indicate which section each table belongs to.

9 The discussion section of the manuscript mentions that probiotics can function by competitively excluding pathogenic bacteria, but no significant effect on the colonization of Campylobacter spp. was observed in the experiment. Does this mean that the mechanism of competitive exclusion is not applicable in this case? It is suggested to further explore why the mechanism of competitive exclusion did not work in this situation.

10 In the discussion section of the manuscript, it is mentioned that probiotics can enhance the host's immune system, but changes in immune organs were only observed at certain time points in the experiment. Does this indicate that the impact of probiotics on the immune system is time-dependent?

11 The discussion section of the manuscript mentions that the in ovo administration of probiotics can improve the intestinal development of broiler chickens, but this improvement only becomes significant at 21 and 35 days after hatching. Does this imply that early feeding has a greater impact on intestinal development?

12 The discussion section of the manuscript mentions that the in ovo administration of probiotics may have a potential impact on the development of the bursa of Fabricius, but the observed effects at certain time points did not align with expectations. Could this discrepancy be due to biases resulting from the experimental design or an insufficient sample size?

13 The discussion section of the manuscript mentions that the in ovo administration of probiotics had no significant effect on the colonization of Campylobacter spp.in the gut, while other studies have shown that probiotics can effectively inhibit the colonization of pathogenic bacteria. Does this suggest that the types of probiotics used or the method of administration in the experiment may not be ideal?

14 The discussion section of the manuscript mentions that the in ovo administration of probiotics has some impact on the production performance of broilers (such as slaughter weight and mortality rate), but this impact is not consistent across different experimental repetitions. Could this be due to differences in environmental factors or experimental conditions?

15 The discussion section of the manuscript mentions that the in ovo administration of probiotics does not significantly affect the hatchability of broiler chickens, but does have a certain promoting effect on later production performance. Does this mean that the effect of probiotics is mainly reflected in the growth stage of broiler chickens, rather than the embryonic development stage?

16 The discussion section of the manuscript mentions that the in ovo administration of probiotics has some impact on the production performance of broilers, but this impact is not consistent across different experimental replications. Please analyze the specific reasons or mechanisms behind this. Does this indicate issues with the reproducibility of the experimental results or suggest potential biases in the experimental design?

17 Please carefully check and adjust the format of the manuscript according to the journal's requirements.

Author Response

Dear Reviewer

We would like to thank you for the revision of our manuscript. We appreciate the effort of the Reviewer. All the comments were carefully considered and all mistakes indicated in the review were corrected according to the Reviewer's suggestions. The detailed responses are provided below. Corrected or added text fragments have been highlighted in the manuscript.

Reviewer 2 Report

Comments and Suggestions for Authors

This manuscript investigates the effects of in ovo administration and early posthatch feeding of three probiotic strains (Bacillus subtilis, Lactobacillus fermentum, and Enterococcus faecium) on intestinal morphology, immune development, and Campylobacter jejuni colonization in broiler chickens. Through histological analysis, immune cell profiling, and microbial quantification at different developmental stages (days 7, 21, and 35), the study aims to evaluate whether these probiotics can enhance gut health and reduce pathogenic bacterial colonization during early growth. The topic is relevant to poultry health management and the development of antibiotic alternatives in livestock production.

Major Comments

  1. Inconsistent Interpretation of Campylobacter Colonization Results

The most critical issue in this manuscript is the inconsistency between the Campylobacter jejuni colonization data and the conclusions drawn. Although probiotics were expected to reduce colonization, the data—especially on day 35—clearly demonstrate higher colonization rates and bacterial loads in the LAC and ENT groups compared to controls. This finding should not be underplayed or generalized as "no reduction"; rather, authors should address the possibility that certain probiotics may even enhance colonization under specific conditions. A more rigorous and cautious interpretation is needed.

  1. Presentation of Supplementary Data

Supplementary Figure 1, which shows Campylobacter CFU counts, is central to the study’s conclusions. This figure should be moved into the main manuscript and more clearly referenced in the Results and Discussion sections. Additionally, Table 1 and Table 2 only list primer sequences; these are methodological details and should be moved to Supplementary Information instead of being presented as main results.

  1. Insufficient Discussion of Probiotic Colonization Data

While the study reports the PCR detection rate of each probiotic strain over time, the relationship between probiotic colonization levels and immune modulation or Campylobacter colonization is not analyzed. Consider correlating probiotic detection rates with key outcomes to reinforce or challenge the study’s assumptions.

  1. Fragmented and Ambiguous Immunological Findings

The effects of probiotics on immune organs (bursa, spleen, cecal tonsil) are not consistent across time points or cell types. For example, the BAC group showed an increase in spleen B cells but a decrease in CD4+ cells. These mixed results require further clarification and should be discussed in the context of existing literature. The biological relevance of these changes is also not sufficiently explored.

  1. Literature Review Needs Strengthening

Although the Introduction provides a general overview of probiotic applications in poultry, it lacks depth regarding time-sensitive effects of probiotics on Campylobacter colonization and potential disruption of microbial homeostasis. Please include recent high-impact studies (e.g., Frontiers in Microbiology, Poultry Science) that discuss the paradoxical outcomes of certain probiotic interventions.

Minor Comments

  1. English Language Needs Polishing

While generally readable, some sentences are overly long and could be more concise. For instance, the sentence in the Introduction starting with "Among the various probiotic strains..." could be rewritten as:

“Among the probiotic strains, B. subtilis, L. fermentum, and E. faecium have demonstrated notable effects in poultry.”

  1. Clarify Statistical Reporting in Figures

Figures (especially Figure 1–3) should clearly indicate sample sizes, statistical tests used, and significance levels (e.g., P < 0.05, P < 0.01). This will improve transparency and allow better assessment of result reliability.

  1. Methods Section Needs Streamlining

Sections 2.2 and 2.7 contain redundant descriptions of DNA extraction and PCR. Please consolidate and simplify these methods to improve readability.

  1. Abstract Should Reflect Key Findings on Campylobacter

The Abstract currently lacks any mention of the Campylobacter findings, which are crucial. Consider adding a sentence such as:

“However, in ovo probiotic administration did not reduce Campylobacter jejuni colonization and even increased bacterial loads in some groups by day 35.”

  1. Graphical Presentation Could Be Improved

Replace line plots with bar charts showing group-wise means ± SD and percentage of positive samples for each group and time point. This will make trends more discernible and interpretation easier for readers.

Overall Recommendation: Major Revision

This manuscript addresses an important topic using a well-designed animal model and multiple outcome measures. However, the conclusions are not adequately supported by the data—particularly regarding the Campylobacter results. Additionally, issues related to data presentation, statistical reporting, and clarity of discussion require substantial revision. With these improvements, the manuscript could make a meaningful contribution to the field.

Author Response

(The authors gave the same response as above.)

Reviewer 3 Report

Comments and Suggestions for Authors

Madej et al sought to determine the effect of in ovo administration of probiotics with early post-hatch feeding on gut carriage of campylobacter, development, and performance. This study is of clear interest to the audience of Microorganisms. The study is well conducted, and only minor revisions are needed. In particular, the authors need to add if in ovo administration caused any reduction in hatch.

Please add a reference for the statement that “B subtilis […] produces enzymes that aid in nutrient digestion.”

Stay consistent with the term microbiota throughout the manuscript. The authors use the word microflora and should remove this term as it refers to plant life, not bacteria.

The authors should cite recent studies on E. faecium as a probiotic in chickens as these studies are highly relevant to the aim of this paper:

Olsen, M., et al. 2022. The effect of a probiotic E. faecium 669 mitigating Salmonella Enteritidis colonization of broiler chickens by improved gut integrity. Poultry Science 101(10): 101029.

Lyte, M., et al. 2025. Use of a microbial endocrinology designed dopamine-producing probiotic to control gut neurochemical levels associated with the development of gut inflammation. Poultry Science 104(5): 105028.

As there are no line numbers in the manuscript, comments are ordered following the section of the manuscript text.

Materials and Methods

What does “prepared and delivered by BIO GEN” mean? Specifically, how was each strain prepared? How delivered? Please add this information to the text.

Why were these three strains (B subtilis, L fermentum, and E faecium) chosen? Add information to Methods

From where were these three strains isolated originally? Add this information to the methods.

Why cultured at 37C? The body temperature of a bird is 41-42C.

“Sequences obtained were compared to sequences from GenBank.” The authors need to add information to the text detailing what percent similarity between obtained sequence and GenBank sequence was used in order to confirm species-level identity.

The authors need to add to the manuscript if the administration of probiotics to the eggs resulted in any embryo death. Did any in ovo administration cause reductions in expected hatch number? This information is critical and needs to be added to the manuscript.

Considering this study centers on in ovo administration of probiotics, the authors need to add substantially more detail regarding egg incubation parameters. What temperatures, humidity, and other factors were maintained during incubation up until hatch?

What type of LED illumination? Wavelength, intensity, etc…

Was a negative (or blank) sample swab collected and analyzed?

What part of the small intestine was a swab sample collected from?

What were the “same environmental conditions” that the birds were raised on? Detail these in the methods section.

What was the stocking density, and litter type (Fresh? Reused?)

Would the authors please add a figure to the manuscript, in which the authors present representative images of the H/E staining, villi/crypt measurements?

In all bar graphs, the authors should add individual data points to show the spread of data.

Author Response

(The authors gave the same response as above.)

Round 2

Reviewer 1 Report

Comments and Suggestions for Authors

The author has answered my question or doubt.